

# Temporal and spatial variability of ice cover occurrence on Carpathian rivers: A regional perspective

Maksymilian Fukś[1] and Łukasz Wiejaczka[2]

[1] Department of Geoenvironmental Research, Institute of Geography and Spatial Organization, Polish Academy of Sciences, Krakow, Poland, ORCID: 0000-0001-5929-9789, e-mail: fuksmaksymilian@twarda.pan.pl

[2] Department of Geoenvironmental Research, Institute of Geography and Spatial Organization, Polish Academy of Sciences, Krakow, Poland, ORCID: 0000-0002-4222-6142, e-mail: wieja@zg.pan.krakow.pl

*Correspondence to*: Maksymilian Fukś (fuksmaksymilian@twarda.pan.pl)

**Abstract.** This article presents an analysis of the temporal and spatial variability of ice cover (IC) occurrence from 1950 to 2020 in the Polish part of the Carpathians, with a focus on climatic conditions and the impact of dam reservoir operations. Data on border ice (BI), total ice cover (TIC), and air and water temperature data were collected and analyzed using

complementary statistical methods, such as Sen's slope, linear least squares regression, the Mann–Kendall test, Student's t-test, the Pettitt test, and the Mann–Whitney U test. Additionally, trends and tendencies across multiple time windows were analyzed through Moving Average and Running Trend Analysis. The study found a decrease in the frequency of IC (the sum of the number of days with BI and TIC) and a transformation in the IC structure characterized by an increase in the number of days with BI and a significant decrease in the number of days with TIC. The results suggest that the observed changes in the ice

regime of Carpathian rivers are primarily driven by warming winter air temperatures and the effects are compounded by reservoir operations, which intensify the climatic changes and significantly reduce IC occurrence downstream of their locations.

**Keywords**

River ice cover, reservoir, dam, climate change, Carpathian Mountains

## 1.   Introduction

River ice cover (IC) is a characteristic natural phenomenon of rivers in temperate and circumpolar climate zones. Its occurrence is important for hydrological and geomorphological processes, among other things, and plays a key role in shaping

the habitat conditions of rivers. IC transforms physical conditions in the riverbed, affecting variables such as water temperature, the amount of incoming solar radiation, water oxygenation, and flow dynamics (Prowse, 2001a; Thellman, et al., 2021). It also significantly impacts erosion, transport, and sediment deposition processes, thereby contributing to changes in river channel morphology (Prowse, 2001a). The presence of IC affects biotic elements of the environment by regulating water temperature and dissolved oxygen levels, which influences river productivity and biodiversity (Prowse, 2001b). River IC is also important

for human activities because of its impact on inland navigation and hydraulic structures (Burrell et al., 2021). Ice jams, a





notable consequence of IC, frequently lead to flooding, which can cause material losses amounting to hundreds of millions of dollars and occasionally result in fatalities (Rokaya et al., 2018). On the other hand, in circumpolar regions, frozen rivers serve as winter transportation corridors and play an important role for local communities by performing an important ecosystem service (Brown et al., 2023).

Recent studies suggest that there has been a decline in the frequency and extent of IC occurrence on a global scale (Yang et al., 2020; Newton and Mullan, 2021), which has resulted in the late formation and early breakup of IC (Fukś, 2023). Within Europe, a decrease in the frequency of river ice events has been reported in the catchments of the Danube (Ionita et al., 2018; Takács et al., 2018), Oder (Marszelewski and Pawłowski, 2019), and Vistula (Szczerbińska, 2023; Kochanek et al., 2024) rivers, among others. The main reason for the change in river ice regimes is the increased air temperature in the autumn-winter-

spring period, which results in the later formation of IC, reduced ice thickness, and an earlier breakup (Prowse et al., 2007; Fukś, 2023). The influence of climate change on river IC occurrence is particularly noticeable in the spring period, where the earlier breakup of the IC has been recorded, but the detailed connections between river flow, air temperature, water temperature, precipitation, and IC breakup are not sufficiently understood (Beltaos and Prowse, 2008; Lesack et al., 2014; Cooley and Pavelsky, 2016).

At local and regional scales, the ice regimes of rivers are also being transformed by human activities, mainly due to the regulation of rivers with hydraulic structures and thermal emissions (Weeks and Dingman, 1972; Takács et al., 2013). Reservoirs play a particularly important role in shaping river ice conditions (Takács et al., 2013; Pawlowski, 2015; Jasek and Pryse-Phillips, 2015; Fukś, 2024; Fukś et al., 2024). The thermal stratification of the water in reservoirs during the winter causes warmer water in rivers downstream from their locations because of the release of warm bottom water and the high

thermal inertia of water (Cai et al. 2018; Kędra and Wiejaczka 2018). Reservoirs also disrupt the natural synchronization between air temperature and river water temperature (Kędra and Wiejaczka, 2016), capture mobile ice forms from the upper catchment area, (Starosolszky 1990), and change the dynamics of river flow (Kędra 2023), all of which affect river ice. This means that the temporal and spatial variability in IC occurrence due to environmental conditions is superimposed on the influences of human activity, which can modify local river ice conditions.

The combined influences of climatic and anthropogenic factors necessitate the inclusion of both factors in regional analyses of changes in river ice regimes. To date, few studies have attempted the detailed assessment of trends in the occurrence of river ice phenomena in the context of both climatic and anthropogenic factors (e.g., Pawlowski et al., 2015; Takács et al., 2015). Typically, river sections with little environmental transformation by humans are selected for studies on the impact of climatic conditions on ice processes (Vuglinsky and Valatin, 2018; Kochanek et al., 2024), while, studies of human influence (e.g.,

operation of hydraulic structures or thermal emissions) on ice usually include single rivers (Takács et al., 2013; Tuo et al., 2018; Apsîte et al., 2016). Additionally, most current research on river ice occurrence trends has focused on large lowland rivers where long measurement series are available. Studies that address changes in the ice regimes of smaller mountain rivers on a regional scale remain scarce. As a result, the typical courses of river ice processes in mountainous areas are poorly understood. The lack of appropriate methods for analyzing and visualizing ice data is also a problem. Due to significant





fluctuations in the annual number of days with IC and the occurrence of years with no river ice events, simple trend analyses based on linear models can lead to misleading conclusions.

There are two main objectives of the research presented in this article: i) to determine the direction and magnitude of long-term trends in the duration and dates of IC formation and disappearance on rivers in the Polish part of the Carpathian Mountains, and ii) to present a methodology for studying the long-term variability of river ice phenomena. To accomplish

these objectives, the study has three specific goals: i) identifying the spatial and temporal variability of IC occurrence in the study area, ii) determining the main causes of the observed changes (i.e., climatic factors vs anthropogenic factors), and iii) identifying the changes in the structure of IC occurrence.

The hypothesis tested in this study is that on a regional scale (i.e., the entire area of the Polish Carpathians) there is a decrease in the frequency of IC (i.e., later IC formation, earlier breakup, and decreased duration) because of increased winter

air temperatures, an effect that is locally exacerbated by the operation of dam reservoirs.

## 2. Datasets and methods
### 2.1 The study area

The Polish part of the Carpathian range has an area of 19 600 km² (the latitudinal extent is approx. 250 km and the

longitudinal extent is approx. 100 km; Fig. 1). This area is drained by rivers that serve as right-bank tributaries for the upper Vistula River (Poland's longest river), including major rivers such as the Soła, Skawa, Raba, Dunajec, Wisłoka, and San. The Carpathians are distinguished by dynamic hydrological processes driven by the interplay between climate, elevation, and relief (Soja, 2002). A characteristic feature of the relief of the Polish Carpathians is a stepped structure that resulted from the complex geological history of the area (Jankowski and Margielewski, 2014). Elevation differences range from 200 to 2500 meters,

which creates layered climate zones and distinctive water regimes (Starkel, 1972). The western Polish Carpathians are characterized by higher absolute and relative altitudes and greater precipitation than the Eastern Carpathians (Wypych et al., 2018).

The main rivers flow south to north and follow the general slope of the terrain. The river network density is high, ranging from 1.3 km/km² in the foothills up to 4 km/km² in the Beskid Mountains (Ziemońska, 1973). The hydrological regime of

Polish Carpathian rivers exhibits significant temporal and spatial variability, with marked differences between the western and eastern sections. Carpathian rivers are fed by precipitation, melting snow cover, and groundwater drainage. Peak flows occur in the winter and the summer (Dynowska, 1971). The steep gradients, poorly permeable substrates, and poor catchment retention capacities in Carpathian rivers mean that runoff is mostly surface-based. As a result, in rainless periods the watercourses experience low water levels (increasingly prolonged in recent years), while in rainy periods there are sudden

large surges (Punzet, 1975), often on a local scale. The climate of the Polish Carpathians is described as transitional between oceanic and continental climate conditions. The area lies on the route of migration and transformation of air masses with very different physical properties, which most often move from the southwest to the northwest throughout the year (Warszyńska, 1995).





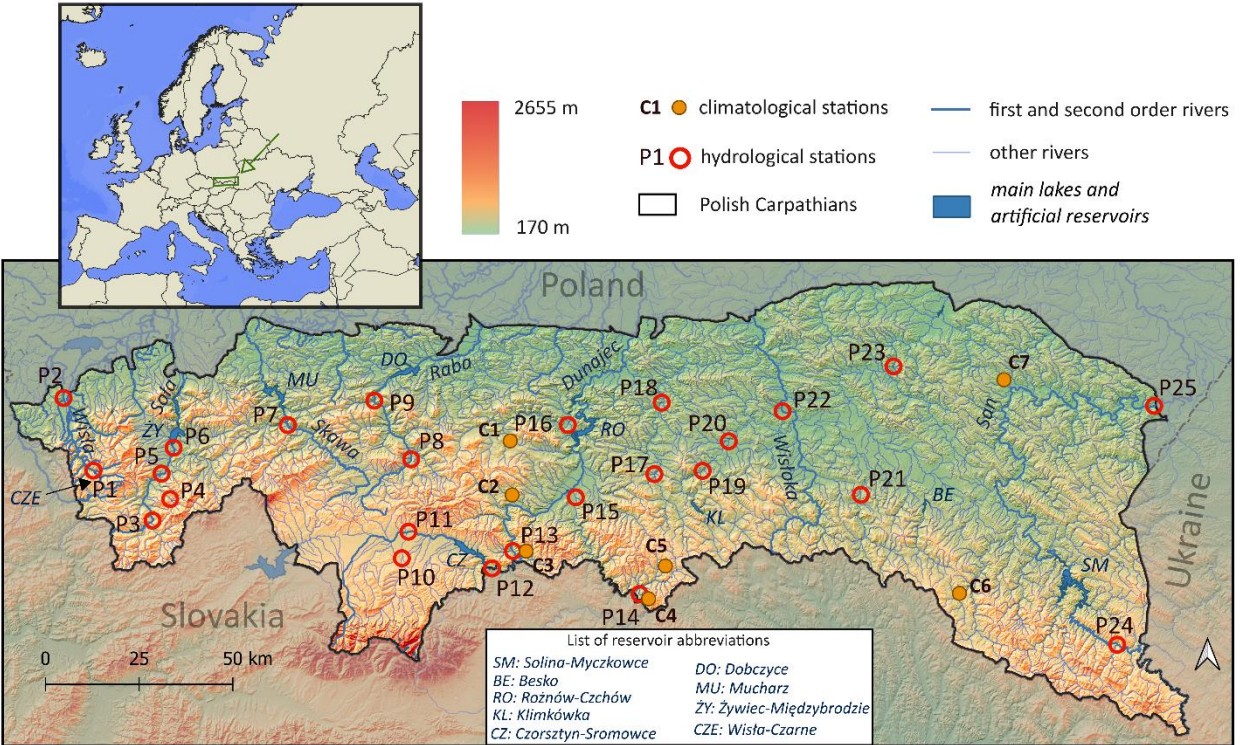

**Figure 1.** Study area and location of hydrological and meteorological stations. NASA Shuttle Radar Topography Mission (2013) data was used to create the map.

The upper sections of Carpathian rivers exhibit minimal anthropogenic transformations. In contrast, the middle and lower reaches have undergone significant alterations due to nearby settlements, which has notably affected environmental features, including water relations. Alterations are related to the long history of regulating river channels in the area (Witkowski, 2021a) and, above all, to the construction and operation of dams and reservoirs (Kędra and Wiejaczka 2018). There are more than a dozen large dam reservoirs operating in the Polish Carpathians with varying parameters, purposes, and ages. The largest reservoir, in terms of capacity, is Solina (472 million m$^3$), while the smallest is Wisła Czarne reservoir (5 million m$^3$). The oldest reservoirs are Porąbka (1936) on the Soła and Rożnów on the Dunajec (1943), and the youngest is Mucharz (2018) on the Skawa. Half of the dam reservoirs function as complexes consisting of two dependent reservoirs (Solina-Myczkowce, Czorsztyn-Sromowce Wyżne, Rożnów-Czchów). The main purposes of the Carpathian reservoirs include: flood mitigation, low-flow augmentation, hydropower generation, flow equalization, and the provision of water for municipal and industrial purposes. Most of the dam reservoirs operating in the Polish Carpathian Mountains (Solina-Myczkowce, Besko, Klimkówka, Mucharz) close the Beskid river basins. Other reservoirs are located within the mid-mountain basins (Czorsztyn-Sromowce Wyżne, Tresna, Porąbka) and the Carpathian foothills (Rożnów-Czchów, Dobczyce). The dams of Carpathian reservoirs are often located in valley gorges.





## 2.2 Data and data quality

The main sources of data on IC occurrence and air temperature in the study area are measurements and observations from
water gauge cross-sections and climatological stations located in the Polish part of the Carpathians (Fig. 1). Data on border ice
(when the water surface is partially occupied by ice; BI), total ice cover (when the entire water surface is occupied by ice;
TIC), and IC (the sum of the number of days with BI and TIC) occurrence at 25 water gauge cross-sections from the period
1950–1980 were obtained from the Hydrological Yearbooks published each year by the Polish Institute of Meteorology and
Water Management - National Research Institute (IMWM-NRI; which was called the National Hydrological and
Meteorological Institute until 1972). For the period 1981–2020, data on IC occurrence was obtained from the IMWM-NRI
online database (https://danepubliczne.imgw.pl/data/dane_pomiarowo_obserwacyjne/). Daily measurements of the occurrence
of different types of ice phenomena (total ice cover, border ice, frazil ice, etc.) and irregular measurements of ice thickness
were conducted at the cross-sections.

BI and TIC were chosen to study the variability in IC occurrence because IC is discontinuous in mountainous rivers with
steep gradients and often occurs in the form of a bridge combined with BI (Beltaos, 2013). In the study area, BI is usually the
first intrinsic stage of the formation of total river IC. The number of days with TIC and BI in the period 1900–1954 was used
to characterize the ice regime in the first half of the 20$^{th}$ century and was determined based on archival data from a book by
Gołek (1957).

Due to the nature of the ice data (the database only contains information on the occurrence or absence of a specific ice
phenomenon on a given day), it is challenging to assess the quality of the acquired data. In the period 1980–2020, ice
information was made available only for days of the year on which some ice phenomenon was recorded. On other days,
IMWM-NRI does not state whether the lack of information is due to the absence of ice phenomena or the lack of observations
on a given day. This causes uncertainty; there is a possibility that there were periods in which IC occurred that were unrecorded
due to a lack of observations. To limit the number of water gauges to those with no uncertainty, the IC database was compared
to the water level database. Water gauges where less than 10% of data contained uncertainty were accepted for analysis.
However, this type of validation was insufficient to completely remove uncertainty because of the possibility of automatic
water level measuring stations in cross-sections. In these cases, the database contains information about the state of water in
the cross-section but there is no information about ice phenomena included. Therefore, information on the time that daily work
was completed by observers at the surveyed water gauges was obtained from IMWM-NRI and compared to the data on ice
occurrences. It was assumed that if an observer was making observations on a given day, then the data on ice phenomena (or
lack thereof) is likely correct. In years where ice phenomena were not recorded in the database, but water level data are
available and observers were confirmed to be working, the value of days without ice data was assumed to be zero. Despite the
detailed analysis of the quality of the acquired data, it should be assumed that there may be small errors in the database, which,
due to the lack of detailed information on the dates of observations, cannot be fully recognized.

For stations not influenced by dam reservoirs, data gaps for several gauging stations were completed with data from the
nearest water gauge. In one case, the data series was created using measurements from two water gauges (1950–1972 and



1973–2020) due to the short distance between the stations and the high correlation between the number of days with IC. Based on the acquired data, the sum of the number of days with IC (the sum of days with TIC and BI) and, separately, the number of days with TIC and BI cover were determined. The dates of the first and last days of IC on the river were also determined.

Average daily air temperature data were obtained from seven IMWM-NRI climatological stations (C1–C7) located in the Polish Carpathian Mountains for the period 1962–2020, and data on average monthly air temperature for the period 1950-2020 were obtained from the high-resolution Climatic Research Unit gridded Time Series (CRU-TS) climate dataset (Harris et al., 2020). Data from the CRU-TS dataset were obtained for all grid pixels within which the analyzed water gauges are located. Information on daily water temperature at two water gauge cross-sections, in periods before and after the construction of dam

reservoirs,            was           obtained           from           the           IMWM-NRI           website (https://danepubliczne.imgw.pl/data/dane_pomiarowo_obserwacyjne/). and the resources of the Institute of Geography and Spatial Organization, Polish Academy of Sciences.

     Due to the significant number of research topics addressed in this article, the main text includes the most important figures necessary to support the conclusions of the research. To keep the results of the study fully transparent, figures related to side

issues and detailed figures related to individual water gauges are presented to readers as supplementary material.

### 2.3  Statistical methods

     Due to the complexity of hydroclimatic data, several complementary statistical methods were used to characterize the multi-year variability and trends in IC parameters on the studied rivers. First, the trend in the annual number of days with IC

(total, border, and their sum) was calculated for the period of 1950–2020. The Theil–Sen estimator (Theil, 1950; Sen, 1968) and linear least squares regression (Mudelsee, 2019) methods were used to determine the direction of change. Linear regression analysis was only used to analyze data that had a normal distribution, and normality was verified using the Shapiro–Wilk test (Shapiro and Wilk, 1965). The non-parametric Mann–Kendall test (Mann, 1945; Kendall, 1975) and Student's t-test (1908) were used to test the statistical significance of trends, and results were assumed to be statistically significant at $p<0.05$. To

identify trends for all analyses presented in the article, the original Mann–Kendall test was used, while a modified test for data affected by autocorrelation (Hamed and Rao, 1998) was also applied to exclude the possibility of determining false-positive trends (data not shown in the article).

     To identify possible points of change in the time series, the nonparametric Pettitt test (1979) was used to determine the points at which the distributions change in the sequence of observations. To better understand the direction of change, the

difference in the average number of days with IC between the two sub-periods was also calculated: 1950–1985 (A) and 1986–2020 (B), according to the formula: $diff. = B - A$, and the statistical significance was assessed using the non-parametric Mann–Whitney U-test (Mann and Whitney, 1947). The null hypothesis ($H_0$) assumed that the distribution of group A is equal to the distribution of group B. The alternative hypothesis ($H_1$) assumed that the distributions were not equal. These methods allowed a preliminary determination of the direction and magnitude of changes in individual ice cover characteristics.



Next, Moving Average and Running Trend Analysis (MARTA) was used to identify the nonlinear variability of the included parameters and the current trends. This method was described by Ranzi et al. (2024) and is based on several previous works (Brunetti et al., 2009; Ranzi et al., 2021). It involves calculating the moving average and trend over all possible sub-periods of the data series (for trends longer than 10 years) and then visualizing the results in the form of triangular graphs (MARTA charts; Ranzi et al., 2024). On the graphs, the vertical axis shows the length of the subperiod for which the mean or

trend was calculated, while the horizontal axis shows the middle year of the subperiod. The Theil–Sen estimator was used to determine the trend, while statistically significant values were determined using the Mann–Kendall test. The presence of statistically significant trends in the upper part of the graphs (which represent longer time windows) indicates the presence of long-term, stable trends in the studied time series. In contrast, statistically significant trends in the lower parts of the graphs (which represent shorter time windows) indicate short-term variability. These charts allow the simultaneous comparison of

fluctuations and trends in the studied variables at different time scales. In this paper, MARTA triangles were created based on the average number of days with the studied ice phenomena (BI, TIC, and IC) from all water gauges (P1–P25). Separate MARTA triangles were created for water gauges with no anthropogenic influence (e.g., unaffected by dam reservoirs or large cities) and for water gauges under the influence of dam reservoirs. A list of all water gauges and their classification as affected or unaffected by anthropogenic influences is presented in the supplementary materials (Tab. S1).

MARTA triangles were also created for the average monthly and winter air temperature (1950–2020) from CRU-TS data (Harris et al., 2020). A detailed analysis of thermal conditions was carried out for the period 1962–2020 using daily average air temperature data from seven climatological stations (C1–C7; Fig. 1). For each winter season (November to March), the number of days with average air temperature below 0°C (NDB(0)) and below -5°C (NDB(-5)) was determined. This parameter made it possible to characterize the number of days on which IC formation (BI and its development into TIC) is possible. The

seasonal sum of degree days with negative air temperature (SNDD) was calculated by taking the sum of all negative values of daily air temperatures during the winter period. The magnitude of the trends for all parameters, during the period 1962–2020, was then calculated using the Theil–Sen estimator, and the statistical significance of the changes was verified using the Mann–Kendall test. The results of the analysis (for both ice and air temperature) are presented in hydrological years that begin on November 1 and end on October 31, which allows each winter season to be covered by one hydrological year.


### 3. Results
### 3.1 Temporal and spatial variability of ice cover occurrence

     At the surveyed stations (P1–P25) during the period 1950–2020, IC lasted for an average of 50 days each year. The number of days with IC consisted of TIC (occurring for 18 days on average) and BI (occurring for 32 days on average). On average,

IC first appeared on rivers on day 44 of the hydrological year (December 14), and disappeared on day 118 of the hydrological year (February 26). In the period 1950–1960 (the period before significant anthropogenic and climatic transformations of river water temperature), the maximum recorded ice thickness on the studied rivers exceeded 50 centimeters. The average annual maximum ice thickness during this period was about 21 centimeters (estimate based on irregular measurements). The data





suggest that during the period 1950–2020, compared to the first half of the 20<sup>th</sup> century, there was relatively little change in
the average annual sum of the number of days with IC (IC). It was estimated that between 1900 and 1954, the total number of
days with IC at the 14 water gauges ranged from 34 to 77 days, with an average of 51 days. Significant differences, however,
were noted in the proportions of BI and TIC. In the period 1950–1954, BI occurred for an average of 14 days, while TIC
occurred for 37 days. Thus, based on these data, it can be assumed that in the period 1950–2020, compared to the period 1900–
1954, there was a significant change in the structure of the IC, characterized by a decrease in the frequency of TIC and an
increase in the frequency of BI. However, it should be noted that estimates of the number of days with IC in the 1900–1954
period were based on a smaller number of water gauges (often located in other locations relative to this study) and data
characterized by observation gaps (Gołek, 1957).

Analyses showed that there was a reduction in the duration of IC on rivers in the study area during the period 1950–2020.
The trend determined using Sen's slope ranged from -10 to 2.4 days per decade, with an average value (for all stations; P1–
P25) of -2.3 days per decade (Tab. 1). The majority of stations (20) showed a decreasing trend in the number of days with IC,
while an increasing trend was recorded in only three stations. Statistically significant trends were found for eight stations, most
of which were located in the southern and western parts of the study area. The magnitude and direction of the trends determined
by Sen's slope are very close to the values determined by least squares linear regression, which strengthens both results. The
decrease in the number of days with IC was also confirmed by the differences between the two analyzed periods: 1950–1985
and 1986–2020 (Tab. 1).

**Table 1.** Trends in the number of days with total ice cover, border ice, and the sum of days with these phenomena (in the
period 1950–2020); the point of change of the central tendency; and the difference in the average number of days with ice
cover between periods B (1986–2020) and A (1950–1985).

| Station code | Trend over the period 1950–2020 [number of days per decade]* | | | | | | Point of shift of central tendency* | | | Difference in averages between periods (B–A)* | | |
|---|---|---|---|---|---|---|---|---|---|---|---|---|
| | Total ice | | Border ice | | Sum | | Total ice | Border ice | Sum | Total ice | Border ice | Sum |
| | Sen. | Lin. | Sen. | Lin. | Sen. | Lin. | | | | | | |
| P1 | **-4.1** | | -0.6 | | **-5.5** | | 1988 | | 1988 | **-17.0** | **-7.0** | **-24.0** |
| P2 | **0.0** | | 0.7 | | -0.9 | | 1988 | | | **-9.0** | 1.2 | -7.8 |
| P3 | **0.0** | | 0.0 | | -1.3 | -1.2 | 2007 | | | -1.3 | -2.7 | -3.9 |
| P4 | **-3.7** | | **4.8** | **3.8** | -2.1 | -2.0 | 1988 | 1976 | | **-21.8** | **14.8** | -7.0 |
| P5 | **0.0** | | 0.7 | 0.7 | -2.2 | -1.9 | 1973 | 1962 | 2007 | **-9.2** | 3.8 | -5.4 |
| P6 | **0.0** | | **-4.0** | **-4.0** | -1.8 | -1.8 | 1978 | 2007 | | **9.3** | **-12.4** | -3.0 |
| P7 | **-1.2** | | **3.8** | | 1.0 | 0.7 | 1998 | 1964 | | **-6.6** | **12.7** | 6.2 |
| P8 | **-0.5** | | -0.9 | -0.4 | **-3.6** | | 1982 | | 2007 | **-10.5** | -1.8 | **-12.3** |
| P9 | **-1.9** | | **2.7** | | 0.0 | | 1988 | 1976 | | **-12.7** | **11.9** | -0.9 |
| P10 | **-3.9** | | 2.0 | | **-3.5** | **-3.3** | 1998 | | 1988 | **-17.7** | 5.5 | **-12.3** |
| P11 | **-0.8** | | **3.8** | **3.8** | -1.2 | -1.3 | 1974 | 1976 | | **-18.5** | 10.6 | -8.0 |





| | | | | | | | | | | | | |
|---|---|---|---|---|---|---|---|---|---|---|---|---|
| P12 | **-3.5** | | **-2.9** | | **-10.0** | | 1981 | 1998 | 1995 | **-30.4** | **-11.0** | **-41.4** |
| P13 | **-0.8** | | 0.0 | | **-7.3** | | 1998 | | 1998 | **-23.0** | -1.5 | **-24.5** |
| P14 | **-5.6** | | **3.1** | | -3.2 | -2.6 | 1998 | 1975 | 1997 | **-24.4** | **11.0** | **-13.4** |
| P15 | 0.0 | | 2.0 | 1.9 | -0.5 | -0.2 | | | | -7.1 | 5.0 | -2.1 |
| P16 | 0.0 | | **2.8** | | 0.6 | 0.4 | | 1990 | | -8.6 | **10.8** | 2.2 |
| P17 | **-2.4** | | 0.9 | 0.8 | -2.7 | -2.2 | 1988 | | | **-13.6** | 5.4 | -8.2 |
| P18 | 0.0 | | **2.9** | | 2.4 | 2.1 | | 1974 | | -0.9 | **8.0** | 7.1 |
| P19 | -2.2 | | -0.8 | | **-4.2** | **-3.8** | 1998 | | 1998 | -12.6 | -2.2 | **-14.9** |
| P20 | **0.0** | | 0.0 | 0.2 | -1.7 | -1.9 | 1988 | | | **-7.7** | 0.2 | -7.4 |
| P21 | **-5.4** | | **3.9** | | -0.8 | -0.8 | 2005 | 1999 | | **-16.2** | **12.6** | -3.6 |
| P22 | **-1.3** | | **5.0** | | -1.3 | -1.0 | 1970 | 1966 | | **-19.1** | **17.0** | -2.1 |
| P23 | **-5.2** | | **3.6** | | **-3.1** | | 1988 | 1984 | | **-30.2** | **17.1** | **-13.1** |
| P24 | **-5.0** | | **5.8** | **5.1** | 0.0 | 0.5 | 2007 | 1979 | | **-19.7** | **18.5** | -1.2 |
| P25 | **-6.8** | | **4.0** | | **-4.6** | | 1980 | 1978 | 1988 | **-32.6** | **12.8** | **-19.8** |
| Mean | -2.2 | | 1.7 | 1.3 | -2.3 | -1.2 | 1989 | 1980 | 1996 | -14.4 | 5.6 | -8.8 |

*The trends were determined using the Theil–Sen estimator (Sen.) and linear regression (Lin.). The results for linear regression were included only if the study group had a normal distribution. For trends, statistically significant values at the p<0.05 level (determined by the Mann–Kendall test for the Theil–Sen estimator and the Student's t-test for linear regression) are indicated in bold. For the point of change of central tendency, only statistically significant results (at p<0.05) determined by the Pettitt test are included. For the results of differences between groups B (1986-2020) and A (1950-1985), statistically significant values (at the p<0.05 level) determined by the Mann–Whitney U-test are indicated in bold.


The average difference was -8.8 days. Statistically significant differences in the distribution between periods, determined by the Mann–Whitney U-test, were recorded at nine stations (the majority of these stations also had a statistically significant trend according to the Mann–Kendall test). The point of change in central tendency, determined by the Pettitt test, varied by station from 1988 to 2007. The gradual decrease in the frequency of IC during the studied period is also marked in the MARTA

triangles (Fig. 2, a1). There was a downward trend in the annual sum of days with IC in the time windows from 20 to 60 years. This trend particularly intensified in the second half of the study period, around 2000.



**Figure 2.** MARTA triangles showing the moving average (a, b, and c) and trend (number of days/decade; a1, b1, and c1) in the average annual number of days (using average values from all P1–P25 stations) with border ice (b and b1), total ice cover (c and c1) and the sum of these phenomena (a and a1). In the graphs of a1, b1, and c1, statistically significant values at the p<0.05 level are marked with larger markers.

The sum of the number of days with IC reflects the variability in the occurrence of the different types of IC (BI and TIC). The study area showed a gradual increase in the number of days with BI, especially in the first half of the study period (Fig. 2, b1). During the period 1950–2020, the trend determined by Sen's slope ranged from -4 to 5.8 days per decade, with an average value from all stations (P1–P25) of 1.7 days per decade (Tab. 1). Statistically significant (mostly increasing) trends occurred at 14 stations. The increase in the number of days with BI was also confirmed by the difference between the two study periods (1950–1985 and 1986–2020), which averaged 5.6 days (Tab. 1). Statistically significant differences in the distribution between the two periods, determined by the Mann–Whitney U-test, were recorded for 14 stations. The specific point of change in central tendency, determined by the Pettitt test, varied by station from 1962 to 2007. A deceleration of the upward trend in the annual





number of days with boundary icing was noted after 2000, where decreasing trends began to dominate, specifically for short time windows (Fig. 2, b1).

The opposite trend was seen for TIC, which exhibited a marked decrease over the entire period studied. For both the trends and the differences between the sub-periods, most of the changes were statistically significant (Tab. 1). This is evidenced by
the slopes estimated using the Theil–Sen estimator (an average of -2.2 days per decade between 1950 and 2020) and the difference between the two sub-periods studied (an average of -14.4 days). The decline in the frequency of TIC over the study period is also shown in the MARTA triangles (Fig. 2, c and c1). Significant downward trends throughout the study period translated into the disappearance of the occurrence of TIC in the study area, especially after 2000.

The results of trend analyses based on Sen's slope suggest that there was a slight trend toward later IC formation and earlier
decay at some of the stations studied; the results are summarized in the supplementary materials (Tab S1. and Fig. S1). The average trend from all stations (P1–P25) suggests IC formation occurs later each decade by an average of 0.28 days per decade and begins to decay earlier by 1.28 days per decade. It should be noted, however, that over the entire period (1950–2020), the identified trends were statistically insignificant for most stations. The MARTA triangles, on the other hand, suggest a later IC freezeup but no clear trends for IC breakup.

In the study area, there was an increase in the frequency of IC from west to east and north to south during the period 1950–2020 (Fig. 3). The increase in the number of days with IC toward the east is associated with an increase in the degree of continentalism, and is likely caused by a decrease in the average temperature of the coldest month in that direction (Barbara Obrębska-Starklowa, 1995). The difference in the average annual amplitude of air temperature between the western and eastern edges of the study area reaches 1.5°C on average, which is reflected in the ice conditions of the rivers. The decrease in the
number of days with IC from south to north is due to the increase in average air temperature as the altitude above sea level decreases (by 0.5°C/100m on average). Declining trends in the annual number of days with IC were recorded in the southern and western parts of the study area, while the water gauges with increasing trends were concentrated in the north. The largest downward trends occurred primarily at cross-sections located downstream of the large reservoirs. It should also be noted that an increase in the frequency of BI and a decrease in the frequency of TIC were primarily recorded at water gauges located in
the southeastern part of the study area, where TIC was a relatively common phenomenon. At water gauges in the warmer, western part of the study area, there was a decrease in the frequency of both BI and TIC (supplementary materials).





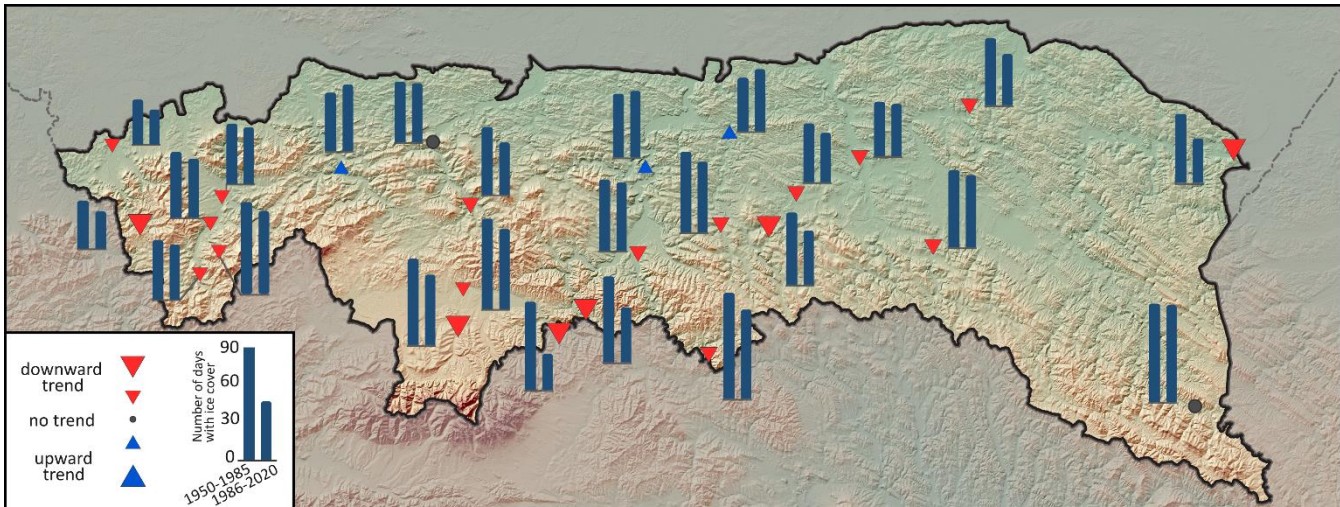

**Figure 3.** Spatial distribution of trends in the number of days with ice cover (1950–2020) and the average number of days with ice cover in the two sub-periods, 1950–1985 and 1986–2020. Values that are statistically significant, as determined by the Mann–Kendall test ($p < 0.05$), are marked with larger markers. NASA Shuttle Radar Topography Mission (2013) data was used to create the map.

It can be concluded that there was a decline in the frequency of IC during the period 1950–2020 that intensified after 2000. Over the same time period, a transformation of the structure of the IC was noted, characterized by a significant decrease in the incidence of TIC and an increase in the incidence of BI. The decrease in the frequency of TIC resulted in the almost complete disappearance of this phenomenon in the study area around 1990, as evidenced by the results of the Pettitt test and MARTA triangles.

### 3.2 Climatic determinants of temporal variation in river ice cover occurrence

To analyze the influence of climatic conditions on the temporal variability of river IC occurrence, MARTA triangles were created based on the average number of days with observed ice phenomena from water gauges unaffected by significant anthropogenic impacts (Fig. 4). It can be assumed that the multi-year variation in the number of days with IC for these water gauges depends mainly on climatic conditions. The analysis showed that these water gauges also experienced a decrease in the frequency of IC but the magnitude of change is much smaller than for all water gauges. Based on Sen's slope, it was determined that there was a decrease in the number of days with IC by 1.1 days per decade for these water gauges over the period 1950–2020. However, most trends were not statistically significant (Tab. 1; Fig. 4, a1). The decrease in the number of days with IC is also suggested by the difference between the two analyzed periods (1950–1985 and 1986–2020). The average difference was -4.24 days, but the change was only statistically significant for four stations (Tab. 1). In contrast, similar to the average of all water gauges (P1–P25), there was an increase in the number of days with BI (2.6 days per decade) and a decrease in the





number of days with TIC (-2.1 days per decade). For these types of ice phenomena, most of the identified trends were statistically significant (Table 1, Fig. 4, b1 and c2).

**Figure 4.** MARTA graphs showing the moving average (a, b, and c) and trend (number of days/decade; a1, b1, and c1) in the average annual number of days at stations (average values from non-human-influenced water gauges) with border ice (b and

b1), total ice cover (c and c1) and the sum of these phenomena (a and a1). In graphs a1, b1, and c1, statistically significant values at the p<0.05 level are marked with larger markers.

The multi-year variability of the number of days with IC (Fig. 4a) coincides relatively well with the variability of the average air temperature of the winter period (Fig. 5a) determined with the CRU-TS dataset. The Pearson correlation coefficient

between these variables was -0.83, which indicates a strong relationship. The high correlation between the average air temperature of the winter period and the number of days with ice partially confirms the relatively good quality of the river ice data. In the study area, the average winter air temperature for the period 1950–2020 (determined using the CRU-TS dataset)





showed a statistically significant upward trend of 0.2°C per decade. The largest, statistically significant trends, which intensified in the second study period, were particularly noted in November and March (Fig. 5c). The period after 2010 was

characterized by especially high air temperature values. The average air temperature of most winters during this period was over 0°C (Fig. 5a), which translated into a small number of days with IC on the studied river sections.

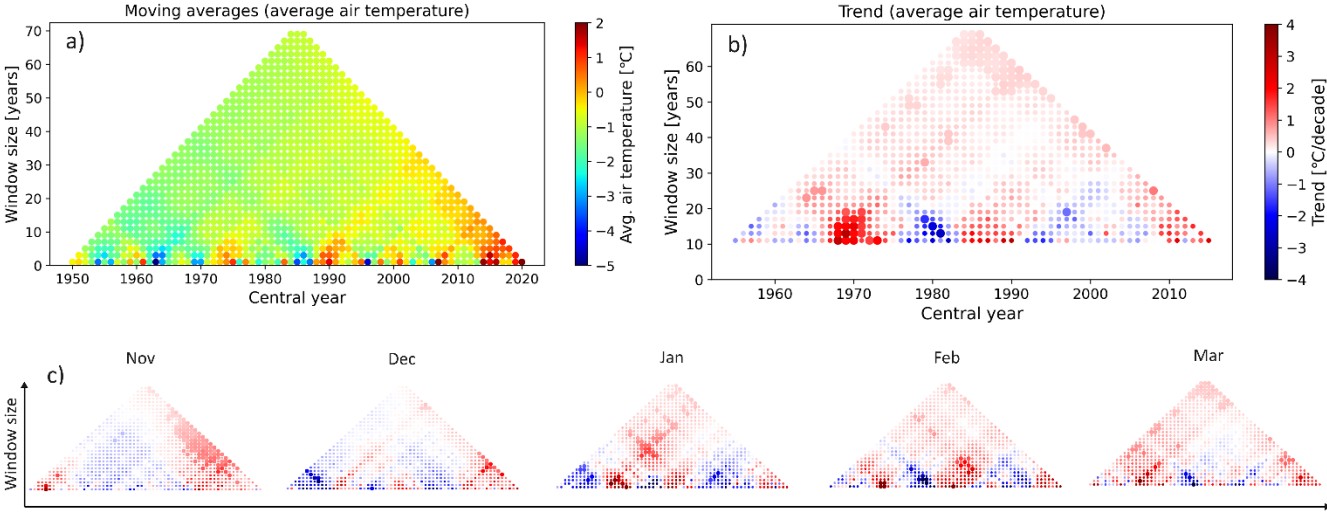

**Figure 5**. MARTA graphs showing the moving averages (a) and trend (b) in the average air temperature of the winter period and individual months (c) for the period 1950–2020. In graphs b and c, statistically significant values at the p<0.05 level are

marked with larger markers.

A detailed analysis of daily data from the water gauge cross-sections from 1962–2020 suggests the occurrence of a gradual warming of the winter period in the study area (Tab. 2). The average winter temperature at the studied stations increased on average by 0.3°C per decade, and the specified trends were statistically significant (at the p<0.05 level). There was a gradual

increase in the seasonal sum of degree days with negative air temperature, averaging 25.8°C·d per decade, which suggests a decrease in the coldness of winters. This trend was statistically significant for the 6 stations analyzed. The number of days with air temperatures below 0°C (NDB(0)) and -5°C (NDB(-5)) also declined at the stations studied. For NDB(0), the decrease was 3.9 days per decade, and the trend was significant for all of the studied stations (Tab. 2). In contrast, for NDB(-5), the decrease averaged 2.5 days per decade, and the trend was significant at six stations.


**Table 2.** Trends in average air temperature of the winter period, seasonal sum of degree days with negative air temperature (SNDD), and annual number of days with average air temperature below 0°C (NDB(0)) and -5°C NDB(-5).

| Indicator | Trend at individual stations (C1-C7) and average trend over the period 1962–2020 | | | | | | | |
|---|---|---|---|---|---|---|---|---|
| | C1 | C2 | C3 | C4 | C5 | C6 | C7 | Mean C1-C7 |





| Mean air temp. [°C/decade] | 0.4 | 0.3 | 0.4 | 0.3 | 0.3 | 0.3 | 0.3 | 0.3 |
|---|---|---|---|---|---|---|---|---|
| SNDD [°C·d/decade] | **28.7** | **23.5** | **33.7** | **26.2** | **24.9** | 23.5 | **24.9** | **25.8** |
| NDB(0) [days/decade] | **-3.6** | **-3.6** | **-4.4** | **-3.2** | **-3.8** | **-3.3** | **-4.3** | **-3.9** |
| NDB(-5) [days/decade] | **-2.8** | **-2.3** | **-3.1** | **-2.5** | **-2.9** | **-2.6** | -2.1 | **-2.5** |

Statistically significant values (p<0.05) are shown in bold.

The results suggest that there was an increase in winter air temperature in the study area, especially in November and March.
The increase in air temperature was accompanied by a decrease in the number of days with an average air temperature below 0°C, which indicates a shorter period during which IC formation is possible, and a decrease in the coldness of winters. The results from CRU-TS data and climatological station data display similar trends in average air temperature, which strengthens our findings. The increase in air temperature is accompanied by a decrease in the frequency of TIC and an increase in the frequency of border ice (BI). In the case of the sum of these phenomena (IC), Sen's slope and MARTA triangles show a
decreasing trend, especially after 2010, but most of the identified trends were statistically insignificant.

### 3.3 Anthropogenic determinants of temporal variation in river ice cover occurrence

To study the influence of reservoirs on IC occurrence within the Polish Carpathians, four cross-sections located downstream of large dams (P1, P12, P13, P19) were analyzed. The analysis showed that these cross-sections had the largest decreases in
the frequency of IC occurrence of all the cross-sections studied. In the period 1950–2020, these stations experienced a decline in the occurrence of IC by an average of 6.7 days per decade, and all trends were statistically significant (Fig. 5). At the same stations, the 1986–2020 period showed a decrease in the number of days with IC by an average of 26.2 days compared to the 1950–1985 period. For all stations downstream of the reservoirs, there was a significant decrease in the frequency of IC in the period after the construction of the dams.
The construction of the Wisła Czarne reservoir (P1; Fig. 1) was completed in 1973. In the 20-year period before the construction of the reservoir, there was an average of 54 days with IC. In the post-construction period, there was an average of 31 days with IC (a decrease of 23 days; Fig. 6). The construction of the Czorsztyn-Sromowce Wyżne reservoir complex (P12 and P13) on the Dunajec River was completed in 1997 (the Sromowce Wyżne reservoir was completed in 1994 and the Czorsztyn reservoir in 1997). At station P12, which is closer to the reservoir, the average annual number of days with IC was
77.9 days for the 20-year period before the construction of this reservoir complex. After construction, it had an annual average of 8.7 days with IC (a decrease of 69.2 days). A slightly smaller impact from the reservoir's operation was observed at station P13, which experienced a decrease of about 45 days after reservoir construction compared to the previous 20-year period. The smaller decrease observed at station P13 can be attributed to the greater distance from the reservoir complex, which results in a smaller change in river water temperature. The construction of the Klimkówka Dam Reservoir (P19) was completed in 1994.





In the 20-year period before the reservoir was constructed, the average number of days with IC was 66.6 days. In the post-construction period, it had an average of 38.6 days with IC (a decrease of 28 days). The dates when the construction of each dam reservoir was completed coincides relatively well with the point of change in central tendency determined by the Pettitt test for stations P12 , P13, and P19 (Tab. 1).

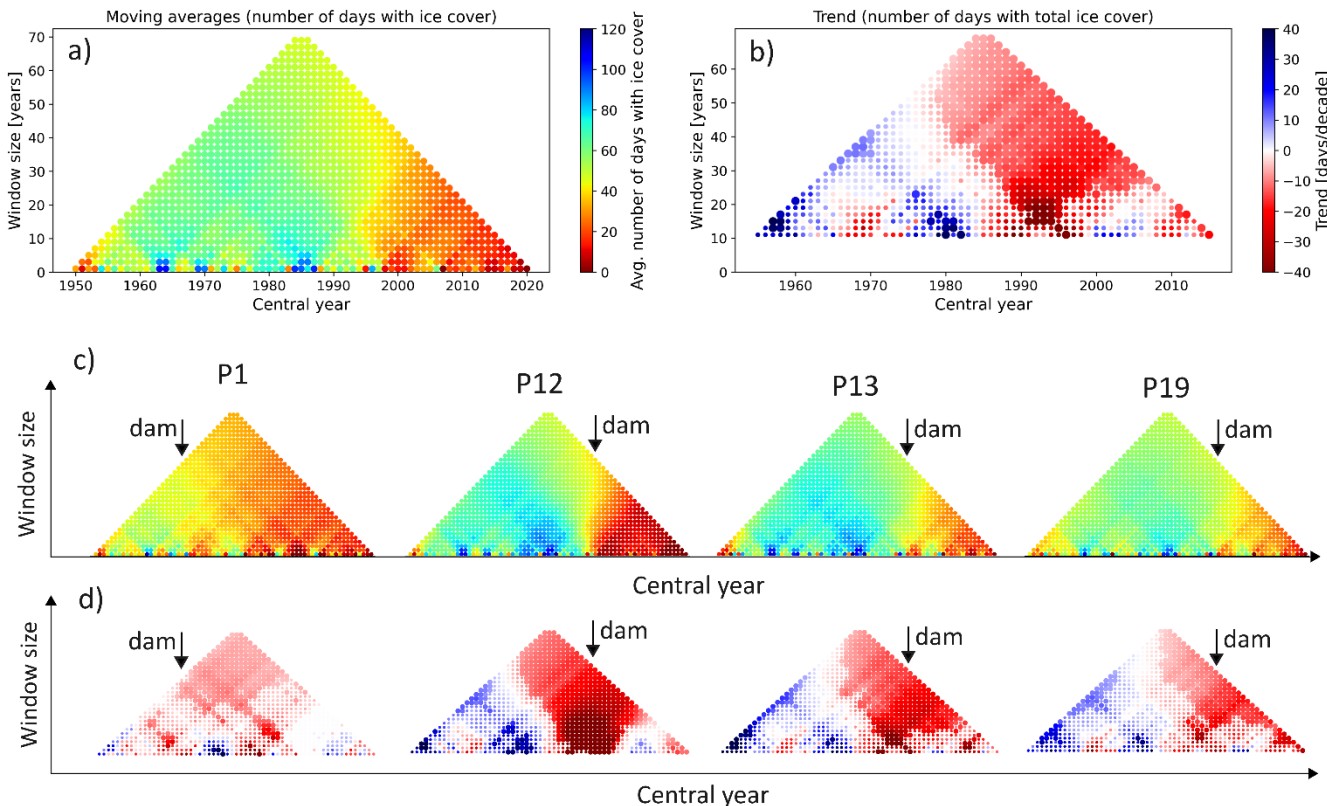

**Figure 6.** Moving average (a) and trend (b) charts showing the number of days with ice cover created from averages of reservoir-influenced water gauges. Panels c and d show MARTA graphs created using data from individual water gauges.

For stations downstream of the reservoirs, the transformation of the ice regime during the study period was significantly greater than the cross-sections influenced only by climatic conditions. The influence of the reservoirs resulted in an almost

complete disappearance of the occurrence of TIC and a significant reduction in the occurrence of BI, which translated into a significant decrease in the frequency of IC. One of the reasons for the reduction in IC occurrence is the increased water temperature below the reservoirs during the winter. The increased water temperature is due to the warmer, hypolimnion bottom waters released by dams. In some areas, this water temperature can reach 4°C, although it is usually 1.5–3°C in Carpathian reservoirs. At station P13 (below the Czorsztyn-Sromowce reservoir complex), an increase of 0.96°C in average winter water

temperature was recorded in the 10-year period after reservoir construction (Fig. 6) compared to the 10-year period before construction. At station P19 (below Klimkówka reservoir), the increase in water temperature was 1.05°C in the 13-year period





after reservoir construction. Analysis of IC occurrence in cross-sections located near large cities (e.g. P25) or in cross-sections located in river basins where there is winter tourist infrastructure such as thermal pools (e.g. P10), suggests that these factors may be of some importance in transforming river ice conditions. These stations showed larger (and statistically significant)
downward trends in the number of days with IC (IC) relative to cross-sections influenced only by climatic conditions.

## 4. Discussion

The Carpathian Mountains are a major biodiversity hotspot in Europe (Bálint et al., 2021). They are characterized by high dynamics of natural processes and are susceptible to the impact of all sorts of external factors (e.g. Werners et al., 2015; Gurung
et al., 2009). The river environments in this region are undergoing significant transformations due to advancing climate change and intense anthropogenic pressures (e.g. Werners et al., 2016; Witkowski, 2021a; Witkowski, 2021b; Wypych and Ustrnul, 2024). Some of the more sensitive elements of the river environment are the thermal and ice regimes of rivers, which are directly influenced by air temperature changes, a primary effect of climate change (Kędra and Wiejaczka, 2018; Fukś et al., 2024). Although temporal changes in river thermal conditions and IC are increasingly studied in case-specific contexts, there
is still a lack of regional perspectives. This is particularly noticeable in the scarcity of studies on temporal-spatial changes in the ice regimes of rivers in mountainous areas.

Studies carried out to date in the area of the Polish part of the Carpathian Mountains, which constitutes a significant part of the upper Vistula River basin, indicate that there has been a gradual decrease in the frequency of ice phenomena in the rivers of this area, for which both climatic conditions and human activity are responsible (Kędra and Wiejaczka, 2018; Szczerbińska,
2023; Fukś, 2023; Fukś et al., 2024). The analysis carried out in this study showed that the river sections unaffected by direct human activity showed a transformation of the structure of ice phenomena that was characterized by a decrease in the frequency of TIC and an increase in the frequency of BI. This suggests that the full cycle of river icing, which in the past characterized the study area (especially the southeastern part) is gradually disappearing. These results were corroborated by an analysis of climatological data, which indicated an increase in average winter air temperature and a decrease in the number of days when
IC formation is possible. It has been shown that in the Carpathians, an increase in air temperature is associated with an increase in river water temperature (Kędra, 2020) and ground temperature (Gądek and Leszkiewicz, 2012), both of which affect IC occurrence. Presumably, an important consequence of the disappearance of TIC is the increased water temperatures in the winter that result from increasing amounts of shortwave radiation reaching the water's surface because of a lack of ice to reflect it.

Our findings are consistent with those reported by other authors, both in central Europe and elsewhere. Kochanek et al. (2024) noted a decrease in the number of days with any river ice phenomena (including ice floes, frazil ice, and ice jams) by an average of 13.3 days per decade in the Carpathian region in the period 1982–2020. Marszelewski and Pawlowski (2019) found a 5.8 days per decade decrease in the frequency of any ice phenomena on the Oder River (Central Europe) and a decrease of 4.6 days per decade for TIC in the period 1956–2015. They also identified an increase in the frequency of years without any
recorded ice phenomena. A gradual decline in IC occurrence on the lower Danube (Central Europe) was described by Ionita





et al. (2018), who calculated that the magnitude of the decline was about 28 days throughout the 20[th] century. They reported an increase in winter air temperature as the reason for the observed changes. For a detailed summary of trends in the duration and timing of IC freezeup and breakup covering many areas of the Earth, see the review article by Fukś (2023). Our findings are consistent with general research on the timing of the appearance and disappearance of ice phenomena. In Europe, IC has been observed to form later and disappear earlier, as documented in the Drava (Takács and Kern 2015), Odra (Marszelewski and Pawlowski 2019), and Torne (Sharma et al. 2016) catchments, among others. It is worth noting, however, that many of the trends we identified in the Polish Carpathians in terms of the timing of IC disappearance were not statistically significant. This may be because our analyses included both TIC and BI. BI often forms during relatively short periods when air temperatures drop below 0°C at the end of winter. On the other hand, the statistically significant trends in the dates of IC appearance are consistent with a significant increase in air temperature at the beginning of winter in the study region, especially in November. Future research should examine the influence of climatic conditions on the occurrence of river IC by undertaking a detailed study of the relationships between the long-term variability of river flow, water temperature, the occurrence of ice phenomena, and the variability of climatic parameters in mountainous areas, which are not fully understood.

In the study area, climate-driven variability is compounded by human activity, particularly the operation of reservoirs. The analysis of ice at water gauge cross-sections located downstream of dams showed that their operation significantly reduces IC occurrence and exacerbates the changes resulting from climatic conditions. These findings are consistent with the results of other studies on the thermal and ice regimes of rivers. Fukś (2024) estimated that there was a decrease in the frequency of IC by 27.3 days in the period after the construction of the Klimkówka reservoir on the Ropa River (Fig. 1, station P19). The effects of the operation of the reservoir were determined to be 77.5% responsible for the difference, and climatic conditions were responsible for 22.5% of the changes. The decrease in IC frequency was associated with a 1.05°C increase in winter water temperature, which was also affected by reservoir operations (65% responsibility) and climatic conditions (35%). Similar results were presented by Wiejaczka (2011) who found that the construction of the reservoir on the Ropa River resulted in a decrease in the frequency of TIC. Cyberska (1972) recorded a significant decrease in the frequency of IC (up to 67%) and an increase in the frequency of ice-free winters after the construction of the Rożnów reservoir on the Dunajec River (1943-1967). A significant (greater than 80%) decrease in ice on the Dunajec River in the period after the construction of the Czorsztyn-Sromowce reservoir complex was reported by Fukś (2024), who estimated that reservoirs may affect IC occurrence for 60 kilometers downstream of the reservoir. The limiting effect of dam reservoirs on IC occurrence has also been reported for rivers in many other areas (Starosolszky, 1990; Belolipetsky and Genova, 1998; Takács et al., 2013; Takács and Kern, 2015; Pawłowski, 2015; Apsîte et al., 2016; Chang et al., 2016; Jasek and Pryse-Phillips, 2015; Maheu et al., 2016). The primary mechanism by which dam reservoirs reduce IC occurrence, especially in mountainous areas, appears to be the warming of downstream river waters during winter (Kędra and Wiejaczka, 2018, Cai et al., 2018, Yang et al., 2022) and the disruption of the natural water–air temperature relationship (Kędra and Wiejaczka, 2016, Fukś et al., 2024). Changing the flow dynamics downstream of the dams and restricting the migration of mobile ice in the longitudinal profile of rivers may also play an important role in IC occurrence (Huokuna et al., 2022). Mobile ice accumulates upstream of the reservoirs and results in the





formation of a compact IC there. The authors' field experience suggests that in the Carpathian region, this process is not analogous to the processes downstream of a reservoir, and the accumulation and formation of the IC upstream of the reservoir affects a smaller area than the area of restricted ice formation downstream.

Archival observational data on river IC can be an interesting and important source of knowledge about changes in fluvial systems due to climate change and human activities. There is some ambiguity about how the different types of IC (TIC and

BI) are recorded. Since the assessments made by the observers are based on visual, subjective observations, the type of ice phenomenon recorded can vary from observer to observer. This problem can be significant in small rivers, where a clear distinction between BI and TIC is not always obvious (Fig. 7).

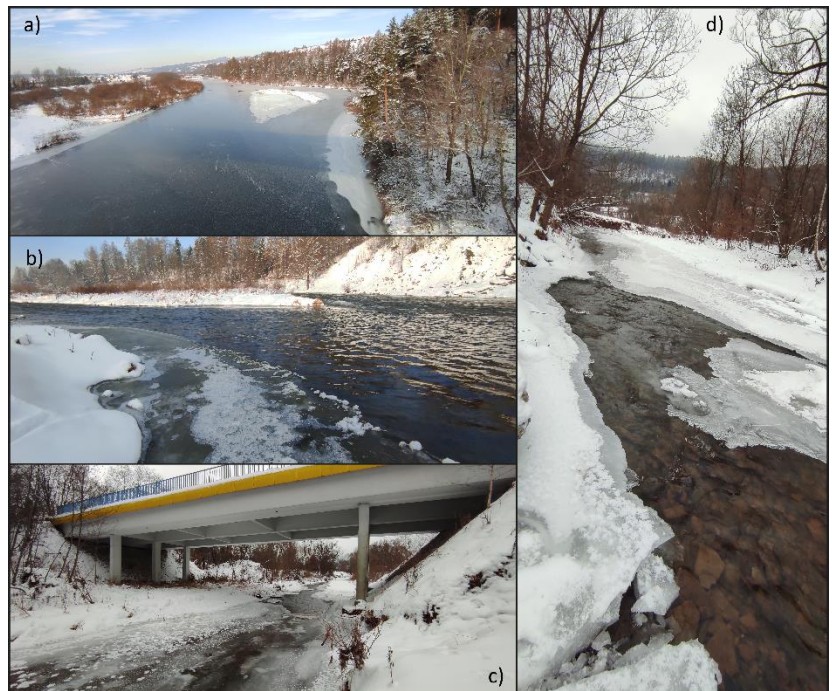

**Figure 7.** Examples of river ice phenomena in the Carpathian region that are unambiguous in their classification: (a) total ice

cover and (b) border ice; and ambiguous classifications (c and d). Photos correspond to the Dunajec (a and b), Ropa (c), and Biala (d) rivers.

Photo: Maksymilian Fukś

However, in the opinion of the authors, this problem does not significantly affect the trend direction of the observed changes in the ice regimes of the studied rivers or the general conclusions of this study, and the collected material is a valuable source

for statistical analysis. It should also be noted that the development of TIC on rivers with different channel widths and flows requires different meteorological conditions. For example, on wide rivers, TIC will take much longer to form than TIC in small mountain streams located at higher relative altitudes.





The analyses showed that MARTA triangles are a useful tool for studying multi-year variability and trends in hydrological (IC occurrence) and climatological (air temperature) data. One of the problems in hydroclimatic time series analysis is the
significant effect that the length of the sub-period has on the presence or absence of the statistical significance of trends in hydrological data series (Burn and Elnur 2002). For example, Fukś (2024) analyzed the variability of winter air temperature at hydrological stations in the Carpathian Mountains from the second half of the 1950s to 2020 and obtained similar trend values to this study (from 0.2°C to 0.4°C per decade) but his results were not statistically significant (with the exception of statistically significant trend in November). Similar results were obtained by Szczerbińska (2023), which also suggested that
trends in air temperature were only statistically significant in November (based on the winter months in the 1981–2018 period). This is because, in addition to the multi-year trend, air temperature displays significant interannual and decadal variability (Kundzewicz, 2017). Consequently, analyses of air temperature trends are often highly sensitive to changes in the start and end dates of the study period. The use of MARTA triangles solves this problem by calculating and visualizing trends for all possible sub-periods, which allows a detailed analysis of the multi-year variability of the studied parameters. It should be noted
that it is necessary to apply an appropriate data pre-whitening method or a modified test to each visualized sub-period when there is strong autocorrelation in hydrological data series (Hamed and Rao 1998). Analyses have also shown the limitations in the applicability of simple trend analysis based on Sen's slope and the Mann–Kendall test when water gauge cross-sections are influenced by reservoirs because the decrease in the incidence of IC occurred rapidly, within the space of a year, and therefore the magnitude of the trend determined using Sen's slope is not reliable. The use of MARTA triangles can better identify points
of rapid change in the IC data series and allows a more effective assessment of the factors affecting river ice.

## 5. Conclusions

This study presents an analysis of the temporal and spatial variability in IC occurrence in the rivers of the Polish part of the Carpathian Mountains (central Europe). Analyses of the temporal variability in air temperature and water temperature in rivers in the study area were also carried out. The conclusions of the research can be summarized as follows:
1. There is a decrease in the frequency of ice cover (defined as the total number of days with border and total ice cover) in the study area that is driven by climate change. However, for the period from 1950 to 2020, most trends are not statistically significant. In water gauges unaffected by substantial anthropogenic impact, the average decline is 1.1 days per decade. MARTA triangles indicate statistically significant trends over shorter periods, with an intensification of the downward trend around 2010.
2. In the study area, for the period 1950–2020, there was a transformation in the ice cover structure characterized by an increase in the number of days with border ice (2.6 days per decade on average) and a significant decrease in the number of days with total ice cover (-2.1 days per decade on average). This indicates the gradual disappearance of the full river ice cycle in the area.

3. The decrease in the frequency of ice cover and the transformation in the structure of ice phenomena are due to the
increase in air temperature during the winter period that averages 0.2°C per decade. The increase in air temperature is





represented by a decrease in the number of days with air temperatures below 0°C (-3.9 days per decade) and an increase in the seasonal sum of degree days with negative air temperatures (an increase in SNDD of 25.8°C·d per decade).

4. In the study area, the multi-year variability of ice cover occurrence is modified by anthropogenic factors. Reservoirs exacerbate the changes from climatic conditions downstream of their location and cause a significant decrease in the frequency of ice cover and the disappearance of total ice cover occurrence. In areas with a large number of reservoirs (such as the Carpathians), their operation can significantly transform the ice regimes of rivers on local and even regional scales.

5. Moving average and running trend analysis (MARTA triangles) is a useful tool to analyze multi-year variability in the number of days with ice cover and other hydroclimatic characteristics. Its use allows a detailed study of the temporal variability of hydrological and climatic phenomena by analyzing trends and moving averages over different time windows.

**Data availability.**

Data on the daily occurrence of ice cover conditions on the studied rivers and daily air and water temperatures at climatological and hydrological stations were obtained from the repository of Polish Institute of Meteorology and Water Management – National Research Institute (IMWM-NRI, 2024, https://danepubliczne.imgw.pl/, access July 1, 2024) and IMWM-NRI surface water hydrological yearbooks (1949–1980). Data on ice and water temperature at station P19 were obtained from the resources of the Institute of Geography and Spatial Organization, Polish Academy of Sciences. Air temperature grid data were obtained from an online dataset (Harris et al., 2020; https://crudata.uea.ac.uk/cru/data/hrg/; access July 1, 2024).

**Author contributions.**

MF and ŁW developed the study concept and prepared the text of the manuscript. MF acquired the data, conducted the analysis, and visualized the results.

Competing interests

The authors declare that they have no conflict of interest.

**Financial Support.**

This research was funded in whole by the National Science Centre, Poland (grant no. 2020/39/O/ST10/00652). For the purpose of Open Access, the author has applied a CC-BY public copyright license to any Author Accepted Manuscript (AAM) version arising from this submission.

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
