# Peer review of "Temporal and spatial variability of ice cover occurrence on Carpathian rivers: A regional perspective"

_Hydrology and Earth System Sciences, 2024_

## Referee Comment (RC2)

[referee-annotated manuscript omitted]

---

## Author Comment (AC1)

Dear Reviewer,

Thank you for your insightful comments and suggestions. The suggestions given to us allowed us to reconsider certain issues and improve the quality of the manuscript.

In the table below, we include all the reviewer's comments and the author's responses.

| Reviewer comment | Authors reply |
|---|---|
| For the data used in the study, have the observers used any guidelines for their registration? E.g. how much border ice must be present before it is recorded? Is total ice cover based on a 100% cover? Is the ice observed in a cross section or over an area? The issue with BI/TIC is mentioned on line 472, is this subjectively evaluated or through any kind of guidelines? | In general, it seems that throughout the period there were no very precise rules for distinguishing ice phenomena in Polish institutions conducting observations. More likely, observers conducted observations according to generally accepted rules in Polish hydrological literature. One such important textbook is "Hydrometry" (eng. Hydrometrics, Bajkiewicz-Grabowska et al., 1993), which states that border ice refers to any occurrence of ice along the shore, while ice cover refers to the total coverage of the water surface by ice. According to this publication, ice phenomena are observed in cross-section.
In addition, in the post-1980 data, the percentage of channel coverage is sometimes given for border ice. Unfortunately, these data are fragmentary and heterogeneous as a result of which their use is problematic.
However, we suppose that the assessment of the occurrence of ice phenomena on the cross sections was to some extent subjective, as we emphasize in the manuscript. This is probably due to the very long tradition of conducting visual observations of river ice phenomena in Poland (the longest series dates back to the 19th century). |
| Did you consider if satellite imagery could be used to verify/check the manual observations just to get some info on the accuracy? | Unfortunately, in the case of Carpathian rivers, this is not possible. We have made some attempts of this type in other studies, but the vast majority of available imagery has too low a resolution to analyze in detail the presence of ice (and BI/TIC distinction) on such narrow rivers. In addition, interpretation is hampered by the presence of islands and various accumulation forms, which are covered with snow and resemble ice.
However, we are now embarking on a study of Europe's larger (wider) rivers, in which satellite data will play a key role. We hope that these studies will shed new light on the quality/detail of these data. |
| In addition to climatic data, ice formation is strongly dependent on river morphology and hydraulics (as you mention in line 481). How similar is your stations? Can different river condition influence the variability between stations? Can you give a brief overview of the river features? | The issue of the influence of channel morphology on the occurrence of ice phenomena is a very interesting and extensive topic that, in our opinion, requires separate, detailed studies based on detailed data and field measurements.
In this article, we will briefly characterize the morphological features of the studied rivers, which will give the reader a better understanding of the conditions in the area. |
| Do you see a change in discharge over time in this region? Could that have an effect on the freeze-up and break-up timing? | The topic of the impact of changes in river discharge on ice cover is a very important one, and we address it as part of our other article which is currently under review. In general, the decrease in IC incidence is strongly correlated with an increase in winter flow. In a significant part of the catchment, an increase in winter flow is observed.
We will address this issue in the text. |
| Can you say something on how much the reservoirs influence the flow? Are the storage capacity of the reservoirs large? From the discussion it seems that it might not be only temperature effects that is influencing the ice but also altered flow dynamics. Some more info on this would be good. | Information will be added to the text (to the discussion and study area chapters) on the effect of reservoir-induced river flow variability on ice occurrence. |
| Regarding the days with no observations (line 142), I assume the ice condition is considered the same until the next observation? I assume this is what is indicated in line 153-154. | We did not assume that an ice phenomenon occurred if it was not clearly indicated in the data series. If the data indicated that there was no ice on a given day, we assumed that the ice phenomenon did not occur. |

| | On the other hand, we focused on excluding all stations where there was an assumption that there were gaps in the observations (see the manuscript for details). If we determined that there were minor gaps in observations, we supplemented the data based on observations from the nearest stations. |
|---|---|
| Line 235: What can cause the increase in IC in some stations? | Our analyses indicate that catchments in the Carpathian region that record an increase in the number of days with ice cover (or no downward trend) also record the absence of a significant increasing trend in winter flow volume. We believe that multi-year changes in flow volume are one of the main factors determining the magnitude of changes in the ice regime of rivers.
The results on the mechanisms of ice regime transformations and its relationship to flow are currently under review in another journal. |
| You see an increase in BI in some stations and a reduction in TIC. It is discussed if the increase in BI is a direct consequence of reduction in full ice cover which sounds reasonable. Could you elaborate on this? It seems like the increasing trend in BI may also indicate a reduced amount of ice. | We agree with the reviewer's comment. It will be added to the text that the decrease in the frequency of TIC and the increase in the frequency of BI indicates an overall reduction in the amount of ice in riverbeds. |
| "dam reservoir" is a special term, wouldn´t just "reservoir" be enough (or dammed reservoir)? | We agree that the word "reservoir" is sufficient. The text will be revised accordingly in terms of this terminology. |
| What is the definition of the hydrological year in Poland? | In the article in the methods section, we pointed out that the hydrological year begins on the first of November and ends on October 31. |
| Line 146-147: "However,…." – I find this sentence difficult to understand, could need some explanation. | The passage will be improved in terms of style. |
| Line 177: should it be normal distribution of residuals? | Of course, linear regression analysis with small samples requires normality of the regression residuals. A corresponding correction will be added to the text. |
| Line 178: Check reference to Student t-test, need an author and not only the year. | The required improvement will be added to the text. |
| Line 183: I assume this means that autocorrelation was no issue? | For the vast majority, the data series over the entire period studied (1950-2020) did not show strong autocorrelation. We checked this using the Ljung-Box test and ACF values. In cases where autocorrelation was found, we checked whether modified tests (several different tests based on variance correction and pre-whitening of the time series) created for analyzing series showing autocorrelation give the same results as the original test. In all cases, the results overlapped. |
| Line 312: Are the significant anthropogenic impacts only reservoir influence? | Of course, the anthropogenic impact can vary greatly, and is very difficult to assess due to lack of data.
In the chapter on the impact of climatic conditions, we limited the analysis to those water gauges not influenced by reservoirs, and also excluded water gauges below large cities and major tourist destinations.
We will include this issue in the text. |
| Line 363: Are the "four cross-sections" here the same as four gauging stations. | Yes, the statement cross sections refers to water gauge stations. It may not be clear in this form, so we will correct it in the text. |
| Line 404: Can you say something more on the external factors | Detail will be added to the text regarding internal and external factors. |

**References**

Bajkiewicz-Grabowska, E., Magnuszewski A., Mikulski, Z.: Hydrometria. Wydawnictwo Naukowe PWN, 1993. [in Polish]

---

## Author Comment (AC2)

Dear Professor Daniele Bocchiola,

Thank you for all your valuable comments and feedback on our manuscript. We agree with most of the suggestions and comments.

Due to the nature of most comments (editorial comments on the text), we do not address all comments separately, but instead post responses to comments where a question was raised or where we ask you to leave the text as is.

| Reviewer comments | Location in text [lines] | Author's reply |
|---|---|---|
| Maybe "it intensifies the impact of climate change"? | 21 | We agree such wording will be much clearer. The text will be corrected this way. |
| - | 129-130, 142-143 | Unfortunately, due to handwriting and clipped letters during the scan, we are unable to read the reviewer's comments. |
| - | 228 | We don't quite understand the reviewer's suggestion. However, we believe that the word "coincides" is appropriate in this context. If it is possible, please leave the text as it is. |
| uphrase this means | 356 | We agree with the reviewer that this statement may be unclear. This sentence will be corrected in the next version of the manuscript by removing the awkward part. |
| I would say "without reservoirs upstream" | 389 | We agree that this is a better statement. The indicated excerpt will be corrected in accordance with the reviewer's recommendations. |
| This is a hypothesis? | 392-393 | This is a fact demonstrated in many scientific studies in the Carpathian region. However, in its current form it is not clear, so we will add relevant citations to the text. |
| Not clear This means presence of geothermal waters? | 398 | We agree that the statement is not clear in this form. The passage refers to geothermal waters, and we will make this clear in the next version of the manuscript. |
| Again "with no tourist infrastructure" | 400 | We agree with the reviewer's opinion, the text will be revised accordingly. |
| How quantified? | 451 | This estimation was based on the XGBoost machine learning model. This information will be added to the text. We believe that due to the extensive methodology, there is no need to provide more details in the text of this article - interested readers are referred to this article by citation. |

---

## Author Response (AR1)

Dear Reviewers and Editor,

Thank you for your valuable comments and suggestions on our manuscript. They allowed us to catch inaccuracies and significantly improve the article from both a substantive and editorial perspective. We believe that the article revised according to your suggestions is much more accessible to readers.

Please find below in the table our responses to all comments. We have also corrected several minor inaccuracies in the text that were not pointed out by the reviewers, but which we noticed at this stage.

For reviewers' comments where no changes were made to the manuscript, we have included the same responses as in the responses to individual comments.

| Reviewer comment | Authors reply | Location of the changes made [lines]* |
|---|---|---|
| **Reviewer 1** | | |
| For the data used in the study, have the observers used any guidelines for their registration? E.g. how much border ice must be present before it is recorded? Is total ice cover based on a 100% cover? Is the ice observed in a cross section or over an area? The issue with BI/TIC is mentioned on line 472, is this subjectively evaluated or through any kind of guidelines? | In general, it seems that throughout the period there were no very precise rules for distinguishing ice phenomena in Polish institutions conducting observations. More likely, observers conducted observations according to generally accepted rules in Polish hydrological literature. One such important textbook is "Hydrometry" (eng. Hydrometrics, Bajkiewicz-Grabowska et al., 1993), which states that border ice refers to any occurrence of ice along the shore, while ice cover refers to the total coverage of the water surface by ice. According to this publication, ice phenomena are observed in cross-section. In addition, in the post-1980 data, the percentage of channel coverage is sometimes given for border ice. Unfortunately, these data are fragmentary and heterogeneous as a result of which their use is problematic. However, we suppose that the assessment of the occurrence of ice phenomena on the cross sections was to some extent subjective, as we emphasize in the manuscript. This is probably due to the very long tradition of conducting visual observations of river ice phenomena in Poland (the longest series dates back to the 19th century). | - |
| Did you consider if satellite imagery could be used to verify/check the manual observations just to get some info on the accuracy? | Unfortunately, in the case of Carpathian rivers, this is not possible. We have made some attempts of this type in other studies, but the vast majority of available imagery has too low a resolution to analyze in detail the presence of ice (and BI/TIC distinction) on such narrow rivers. In addition, interpretation is hampered by the presence of islands and various accumulation forms, which are covered with snow and resemble ice. However, we are now embarking on a study of Europe's larger (wider) rivers, in which satellite data will play a key role. We hope that these studies will shed new light on the quality/detail of these data. | - |
| In addition to climatic data, ice formation is strongly dependent on river morphology and hydraulics (as you mention in line 481). How similar is your stations? Can different river condition influence the variability between stations? Can you give a brief overview of the river features? | We have added a brief description of these features to the article. However, a detailed description of all the morphological and hydrological differences between all the water gauges, and their impact on icing, requires additional scientific research based on a separate methodology. | 97-103 |
| Do you see a change in discharge over time in this region? Could that have an effect on the freeze-up and break-up timing? | Meanwhile, we have published an article on the mechanisms of ice regime changes in Carpathian rivers (including flow changes). We have added a citation and an excerpt referring to this article to the text. | 463-467 |
| Can you say something on how much the reservoirs influence the flow? Are the storage capacity of the reservoirs large? From the | In the discussion, we elaborated on the flow changes caused by reservoirs and their possible impact on icing. | 483-497 |

| | | |
|---|---|---|
| discussion it seems that it might not be only temperature effects that is influencing the ice but also altered flow dynamics. Some more info on this would be good. | | |
| Regarding the days with no observations (line 142), I assume the ice condition is considered the same until the next observation? I assume this is what is indicated in line 153-154. | We did not assume that an ice phenomenon occurred if it was not clearly indicated in the data series. If the data indicated that there was no ice on a given day, we assumed that the ice phenomenon did not occur.
On the other hand, we focused on excluding all stations where there was an assumption that there were gaps in the observations (see the manuscript for details). If we determined that there were minor gaps in observations, we supplemented the data based on observations from the nearest stations. | - |
| Line 235: What can cause the increase in IC in some stations? | Most likely, the slight increase was due to changes in flow volume different from those at the other cross sections. In another of our studies (Fukś and Wiejaczka, 2025), the results suggest that there was no concomitant increase in flow volume at stations where an increase in ice was observed. | 463-467 |
| You see an increase in BI in some stations and a reduction in TIC. It is discussed if the increase in BI is a direct consequence of reduction in full ice cover which sounds reasonable. Could you elaborate on this? It seems like the increasing trend in BI may also indicate a reduced amount of ice. | We have added a reference to this issue in the text. | 434-436, 551 |
| "dam reservoir" is a special term, wouldn´t just "reservoir" be enough (or dammed reservoir)? | The text has been corrected in this regard in many places. | many places in the text |
| What is the definition of the hydrological year in Poland? | In the article in the methods section, we pointed out that the hydrological year begins on the first of November and ends on October 31. | - |
| Line 146-147: "However,…." – I find this sentence difficult to understand, could need some explanation. | The indicated passage has been corrected. | 152-153 |
| Line 177: should it be normal distribution of residuals? | In the previous version, we incorrectly included the distribution in the data rather than the distribution of the regression residuals, which was rightly noted by the reviewer. The analysis has been corrected for this, and the results have been revised in several places in Tables 1 and S1. | 185-186, Table 1, Table S1 |
| Line 178: Check reference to Student t-test, need an author and not only the year. | The text has been corrected. | 189 |
| Line 183: I assume this means that autocorrelation was no issue? | For the vast majority, the data series over the entire period studied (1950-2020) did not show strong autocorrelation. We checked this using the Ljung-Box test and ACF values. In cases where autocorrelation was found, we checked whether modified tests (several different tests based on variance correction and pre-whitening of the time series) created for analyzing series showing autocorrelation give the same results as the original test. In all cases, the results overlapped. | - |
| Line 312: Are the significant anthropogenic impacts only reservoir influence? | In the analysis of the influence of climatic conditions on the occurrence of ice, water gauges were included not only without the influence of reservoirs but also rejected water gauges below large cities and areas developed for tourism. We have included details in the supplementary materials. | - |
| Line 363: Are the "four cross-sections" here the same as four gauging stations. | Yes, the relevant improvements have been added to the text. | 374 |
| Line 404: Can you say something more on the external factors | The text has been revised to include this information. | 419-421 |

| Reviewer 2 – Professor Daniele Bocchiola | | |
|---|---|---|
| Maybe "it intensifies the impact of climate change"? | The text has been corrected. | 21 |
| uphrase this means | The text has been corrected. | 367 |
| I would say "without reservoirs upstream" | The text has been corrected. | 399-400 |
| This is a hypothesis? | This is a fact demonstrated in many scientific studies in the Carpathian region.
Relevant citations confirming this are present in the discussion section. | - |
| Not clear
This means presence of geothermal waters? | The text has been corrected. | 409-411 |
| Again "with no tourist infrastructure" | The text has been corrected. | 414-415 |
| How quantified? | This estimation was based on the XGBoost machine learning model. We believe that due to the extensive methodology, there is no need to provide more details in the text of this article - interested readers are referred to this article by citation. | - |

\* The lines refer to places in the manuscript in change tracking mode.

---

## Author Response (AR2)

Dear Editor,

Thank you for accepting the article for publication. I have not made any changes to the submitted files since the last review, during which all modifications were described in my previous response.

Yours sincerely,

Maksymilian Fukś